

# Injecting structure-aware insights for the learning of RNA sequence representations to identify m6A modification sites

Yue Yu, Shuang Xiang and Minghao Wu

Changjiang Water Resources and Hydropower Development Group, Wuhan, China

## ABSTRACT

N6-methyladenosine (m6A) represents one of the most prevalent methylation modifications in eukaryotes and it is crucial to accurately identify its modification sites on RNA sequences. Traditional machine learning based approaches to m6A modification site identification primarily focus on RNA sequence data but often incorporate additional biological domain knowledge and rely on manually crafted features. These methods typically overlook the structural insights inherent in RNA sequences. To address this limitation, we propose M6A-SAI, an advanced predictor for RNA m6A modifications. M6A-SAI leverages a transformer-based deep learning framework to integrate structure-aware insights into sequence representation learning, thereby enhancing the precision of m6A modification site identification. The core innovation of M6A-SAI lies in its ability to incorporate structural information through a multi-step process: initially, the model utilizes a Transformer encoder to learn RNA sequence representations. It then constructs a similarity graph based on Manhattan distance to capture sequence correlations. To address the limitations of the smooth similarity graph, M6A-SAI integrates a structure-aware optimization block, which refines the graph by defining anchor sets and generating an awareness graph through PageRank. Following this, M6A-SAI employs a self-correlation fusion graph convolution framework to merge information from both the similarity and awareness graphs, thus producing enriched sequence representations. Finally, a support vector machine is utilized for classifying these representations. Experimental results validate that M6A-SAI substantially improves the recognition of m6A modification sites by incorporating structure-aware insights, demonstrating its efficacy as a robust method for identifying RNA m6A modification sites.

# INTRODUCTION

Among the more than 160 discovered methylation types, N6-methyladenosine (m6A) stands out as the most widespread and crucial one in eukaryotes (*Rehman et al., 2021*). This modification entails adding a methyl group to a specific adenine base on the RNA molecule (*Wang & Yan, 2018*). m6A modification plays a pivotal regulatory role in RNA biology (*Fustin et al., 2013*; *Wang et al., 2014*, *2018*; *Geula et al., 2015*; *Wang & Wang, 2020*). Primarily, it is integral to the transcriptional regulation of RNA, exerting influence

Corresponding author
Minghao Wu, wu.mh@crhdc.com.cn

on RNA stability, transport, translation, and degradation rate. During transcription, m6A modifications can be "written" (methylated) into RNA molecules by a group of protein-coding enzymes. Subsequently, a multitude of "reader" proteins recognize the m6A modification, thereby governing the subsequent fate of the RNA. Moreover, abnormal changes in m6A modification are implicated in various diseases, correlating with the onset, progression, and treatment response of these diseases (*Rehman et al., 2021*).

Given the significance of RNA m6A modification, numerous studies have employed high-throughput sequencing methods to identify potential m6A modification sites, delving into the impact of RNA methylation on life processes. These methods encompass MeRIP-Seq, m6A-Seq, PA-m6A-Seq, m6A-LAIC-Seq, and m6A-REF-Seq (*Dominissini et al., 2012*; *Meyer et al., 2012*; *Ke et al., 2015*). They involve segregating m6A-modified RNA fragments from unmodified ones using specific m6A antibodies or chemical reactions, followed by pinpointing the location of m6A modification through high-throughput sequencing technology. These methods not only furnish genome-wide m6A modification site information but also facilitate further exploration of the role of m6A in RNA function and regulation. Nevertheless, such time-intensive approaches grapple with the limitation of high investment and low generalization. To overcome this constraint, an increasing number of studies are turning to computational methods for identifying potential m6A modification sites (*Chen et al., 2015*).

The computational identification of RNA m6A modification sites can be broadly categorized into two groups: machine learning-based methods and deep learning-based methods. Machine learning methods typically rely on hand-crafted sequence features to classify RNA sequences. This approach involves two independent steps: the feature learning process and the feature classification process. It allows the flexible introduction of biological domain knowledge related to RNA sequences, such as the physical and chemical properties of nucleotides. This non-end-to-end approach enables the model to focus on the compositional properties of RNA sequences, including k-mer, one-hot encoding, cumulative nucleotide frequency, composition of k-space nucleic acid pairs, dinucleotide composition, and enhanced nucleic acid composition (*Di Giallonardo et al., 2017*; *Huang et al., 2018*; *Zhang et al., 2020*). Examples of machine learning algorithms in this category include the iRNA toolkit (*Qiu et al., 2017*; *Chen et al., 2018*; *Yang et al., 2018*), AthMethPre (*Xiang et al., 2016*), M6ATH (*Chen et al., 2016*), SRAMP (*Zhou et al., 2016*), and M6AMRFS (*Qiang et al., 2018*). However, this approach faces challenges related to over-reliance on additional information. Additionally, the recognition results of m6A modification sites may be classifier-sensitive, meaning that the choice of different classifiers can affect the overall performance of the model. Commonly used classifiers include support vector machines and random forests. On the other hand, deep learning methods treat feature extraction and classification tasks as continuous processes and are better equipped to extract deep features of sequences. An increasing number of studies are adopting deep learning strategies for RNA m6A modification site identification. For instance, iN6-Methyl (*Nazari et al., 2019*) and m6AGE (*Wang et al., 2021*) utilize convolutional neural networks to extract sequence features.

The methods discussed above primarily focus on the properties of RNA sequences. However, they often overlook the potential structural insights embedded in RNA sequences. Typically, these methods treat RNA sequences akin to natural language, learning the semantic information based on the frequency and position of nucleotides. What distinguishes RNA sequences from traditional natural language is the substantial structural similarities inherent in RNA sequences. Yet, encoding such structural information directly poses challenges. In response, we propose an approach that involves constructing inter-sequence graphs, leveraging the similarity between RNA sequences. By concentrating on the topology of these graphs, we aim to implicitly exploit the underlying structural information present in RNA sequences.

Here, we propose an m6A modification site predictor, called M6A-SAI, which introduces a transformer-based deep learning framework to inject structure-aware insight into sequence representation learning, promoting the accurate identification of M6A modification sites. M6A-SAI uniquely extends the Transformer encoder's capabilities by integrating nucleotide type and position information into low-dimensional representations, which are then used to construct and refine similarity and awareness graphs. Specifically, M6A-SAI first initializes each RNA sequence into a low-dimensional representation containing nucleotide type and position information based on the Transformer encoder (*Vaswani et al., 2017*). Combined with Manhattan distance, M6A-SAI calculates the similarity between sequences based on the representations to construct a similarity graph by exploiting the potential structural differences between sequences. However, the graph based on similarity computation suffer from potential smoothness limitations. To this end, M6A-SAI introduces a structure-aware optimization block to differentiate the topology information of the similarity graph by focusing on key local regions of the graph. This module determines the key topological information of the graph by defining the anchor sets, and further calculates the shortest path distance (SPD) embedding between all sequence nodes and the anchor sets to endow nodes with a certain structure-aware ability. To further enhance the structure-aware information, M6A-SAI uses PageRank to calculate the dissimilarity of each node and obtain the dissimilarity embedding of the solution points. Based on the SPD embedding and difference embedding, M6A-SAI computes the awareness graph of the input sequences. Then, M6A-SAI designs a self-correlation fusion graph convolution framework to integrate the information from the similarity graph and the awareness graph to obtain the final representations of RNA sequences. Finally, the sequence representations are classified using support vector machines. The experimental results show that M6A-SAI significantly enhances the recognition ability of RNA sequence representation by incorporating structure-aware information, and it is an effective method to identify RNA m6A modification sites.

The main contributions of this article are as follows: (1) M6A-SAI incorporates structure-aware information into RNA sequence representation learning by introducing a structure-aware optimization block and awareness graph, enhancing the precision of m6A modification site identification; (2) M6A-SAI constructs a similarity graph using Manhattan distance and addresses smoothness limitations with an optimization block that

focuses on key topological information, thus improving the representation of structural differences; (3) M6A-SAI utilizes a self-correlation fusion graph convolution framework to integrate information from both the similarity and awareness graphs, resulting in more accurate RNA sequence representations for classification. The rest of this article is organized as follows. In the Materials and Methods, we first formally define the problem of RNA m6A modification site identification, and then describe the details of our model. Experimental results are presented in the Results, following which we end this article with a conclusion.

## MATERIALS AND METHODS

### Problem description and datasets

The task of identifying m6A modification sites is conventionally formulated as a binary classification problem. Each sample in a dataset represents a segment of the RNA sequence encompassing the modified central adenine and its adjacent nucleotides. Formally, the objective is to construct and train a model to map RNA sequences to a label set Y = {y}, where $y \in \{0, 1\}$. The fundamental objective is to effectively encode RNA sequences. This model takes the intelligible encoding of RNA sequences as input and strives to accurately assign labels to these representations, optimizing the binary cross-entropy loss function.

In this study, we leverage RNA sequence samples obtained from datasets of three distinct species as benchmark datasets for this classification task. Specifically, we choose the RNA m6A modification site dataset of *Arabidopsis thaliana* (A101 dataset) (*Wan et al., 2015*) for our investigation, comprising 2,100 samples with a balanced ratio of positive and negative samples set. To mitigate the impact of sample imbalance, we used a 1:1 balance of positive and negative samples, ensuring that the model learns equally from both classes, thereby enhancing its generalization ability. Negative samples were selected from the same gene regions as the positive samples to ensure biological consistency in the training data. In addition, negative samples were randomly selected from the same gene regions as the positive samples and are not required to conform to the DRACH motif. The A101 dataset consists of RNA sequences with a length of 101 nucleotides. This length provides sufficient coverage of the upstream and downstream sequence information around the m6A modification site while considering computational complexity. Additionally, we introduced two independent datasets, human-liver and mouse-heart, to further evaluate our model. The human-liver dataset contains 2,634 positive samples and 2,634 negative samples, while the mouse-heart dataset includes 2,201 positive samples and 2,201 negative samples. In both datasets, the sequence length is 41 nucleotides. The sequence lengths of the datasets are consistent with those used in previous studies (*Dao et al., 2020*; *Rehman, Tayara & Chong, 2022*), and all negative samples across the datasets are constructed from a central adenine and flanking nucleotides of equal length. This length is selected based on preliminary analysis and proven effective in capturing local contextual information while minimizing noise and redundancy from longer sequences. This approach generally helps to improve dataset quality by: 1) enhancing the effectiveness and reliability of computational recognition, validating the model's ability to learn latent features for modification site identification; 2) strengthening the model's generalizability, improving

its applicability and robustness across different species, tissues, and experimental conditions.

## Model description

M6A-SAI consists of three integral components that work in tandem to enhance m6A modification site prediction. The process begins with sequence representation initialization, where RNA sequences are encoded into embeddings using a Transformer encoder, capturing nucleotide types and their positional relationships. These embeddings are then used to construct a similarity graph, which is refined by the structure-aware optimization block to produce an optimized awareness graph, addressing limitations in the initial graph by focusing on key local regions and incorporating structural insights. Finally, the self-correlation fusion graph convolution framework processes both the similarity and awareness graphs through distinct graph convolution blocks to generate separate low-dimensional representations, which are then integrated by a fusion block to combine local and global perspectives, resulting in the final, comprehensive RNA sequence representations. The overall architecture and its components are illustrated in Fig. 1.

## Sequence representation initialization

To capture more nuanced associations between RNA sequences, we integrate Transformer-based encoding with Manhattan distance metrics to construct a similarity graph that reflects the semantic relationships inherent in RNA sequences. Within the Transformer encoder framework, two core components are utilized: the positional encoding block, which incorporates positional information to differentiate nucleotide locations, and the self-attention mechanism, which enables the model to weigh and integrate dependencies between nucleotides across the sequence. Additionally, to delve deeper into the structural relationships within RNA sequences, we incorporate the calculation of Manhattan distance between vector representations of the output from the Transformer encoding block. Moreover, we rigorously constrain the values of the structure matrix within the set {0, 1}. The details of these components are outlined below.

Position encoding of Transformer is a functional encoder, that is, the position vector $\mathbf{p} = [p_i]$ can be computed as:

$$p_i = \begin{cases} \sin(w_i \cdot t) & i = 2k \\ \cos(w_i \cdot t) & i = 2k+1 \end{cases} \tag{1}$$

where $w_i$ denotes the frequency, which is calculated as follows:

$$w_i = \frac{1}{m^{2i/10,000}} \tag{2}$$

where $m$ is the number of sequences. Thus, the RNA sequences are represented by K-mer and position encoding. We obtain the sequence representation matrix $\mathbf{R}_s = [\mathbf{r_j}] \in \mathbb{R}^{n \times d}$, where $\mathbf{r_j}$ can be computed as:

$$\mathbf{r}_j = \mathbf{r}_{k-mer} + \mathbf{p} \tag{3}$$

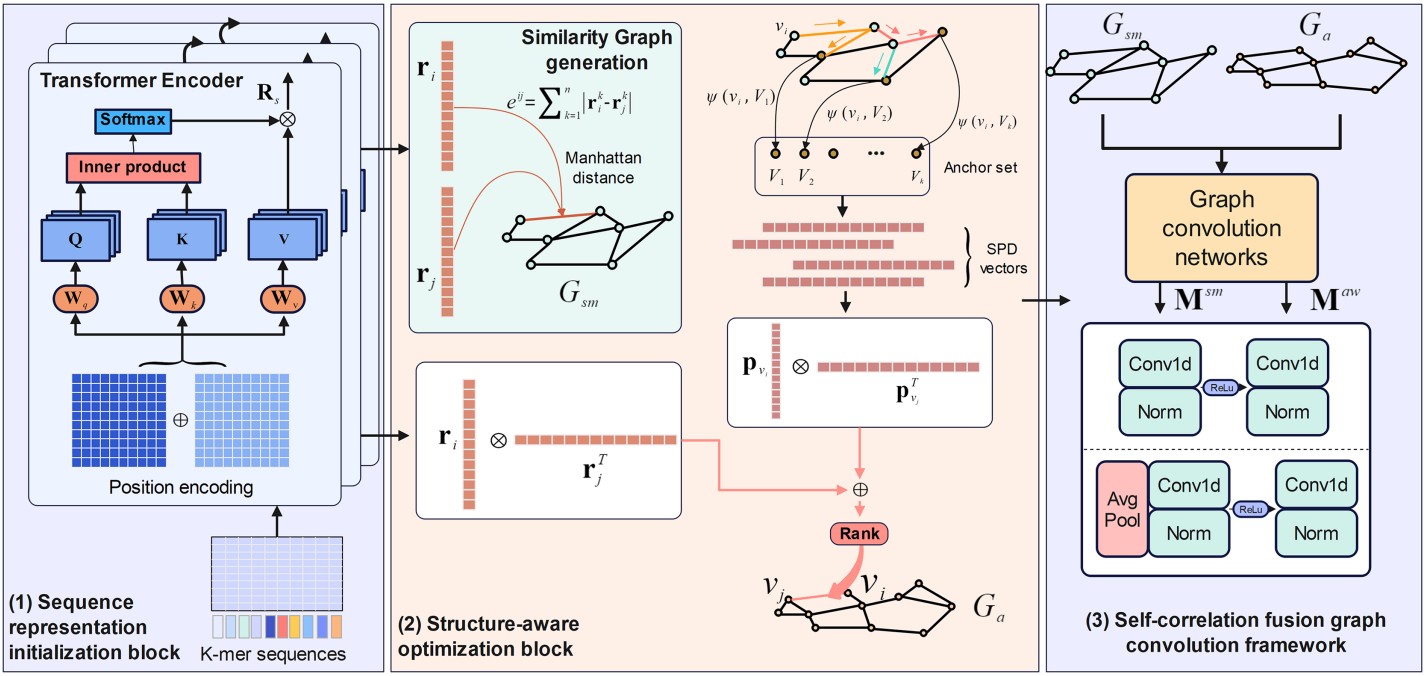

**Figure 1** The framework of M6A-SAI.

where, $\mathbf{r}_{k-mer}$ denotes the K-mer representations. After computing sequence representations, the encoder further optimizes the representation matrix $\mathbf{R}_s$ through the self-attention mechanism:

$$\begin{cases} \mathbf{Q} = \mathbf{W}_q\mathbf{R}_s \\ \mathbf{K} = \mathbf{W}_k\mathbf{R}_s \\ \mathbf{V} = \mathbf{W}_v\mathbf{R}_s \\ \mathbf{R}_s = \text{softmax}\left(\dfrac{\mathbf{Q}\mathbf{K}^T}{\sqrt{d}}\right)\mathbf{V} \end{cases} \tag{4}$$

where, $\mathbf{Q}$, $\mathbf{K}$ and $\mathbf{V}$ are the query matrix, key matrix and value matrix respectively. softmax(·) denotes the *softmax* activation function. Through the self-attention mechanism, the Encoder can reveal the potential associations within the sequence and further excavate the structural associations of nucleotides.

## Structure-aware optimization block

We believe that it is a valid way to describe the associations of RNA sequences through internal relationships captured by the self-attention mechanism. To this end, we also measure this structural association by the Manhattan distance and construct the similarity graph $G_{sm} = (V, E)$ for input sequences, where $V$ is the node set and E is the edge set. For an arbitrary edge $e_{ij}$ of node $v_i$ and $v_j$, we determine it as:

$$e_{ij} = \sum_{k=1}^{n}\left|\mathbf{r}_i^k - \mathbf{r}_j^k\right|. \tag{5}$$

In addition, we ensure that the values of elements in $G_{sm}$ are strictly constrained in the set of $\{0, 1\}$ by threshold filtering, as shown below:

$$e_{ij} = \begin{cases} 1 & if \ e_{ij} \leq 0.5 \\ 0 & otherwise. \end{cases} \tag{6}$$

The similarity graph constructed using Manhattan distance may suffer from excessive smoothing, even with the application of threshold filtering to alleviate this issue. To address this limitation, we propose the integration of structure-aware blocks that are specifically designed to construct a perception graph. This approach enhances the similarity graph by incorporating constraints that emphasize local topological features, thereby improving the representation of structural nuances within the graph. The structure-aware block comprises three components: shortest path distance encoding and perceptual graph construction. Shortest path distance encoding is employed to learn potential global topology information of nodes within the similarity graph. The construction of the perception graph takes into account the importance of nodes to ascertain potential relationships between nodes in the perception graph. The specific descriptions of these two components are detailed as follows.

For an undirected graph, the shortest path distance (SPD) is one of the main ways to measure its topological properties. To this end, for a given similarity graph, we randomly select K nodes to construct the anchor set $\{\mathcal{V}_1, \mathcal{V}_2, \ldots, \mathcal{V}_K\}$. We further compute the SPD vector between all nodes in the similarity graph and this anchor set. For a given node $\mathbf{v}_i$, its SPD vector can be computed as:

$$\mathbf{p}_{v_i} = (\psi(v_i, \mathcal{V}_1), \psi(v_i, \mathcal{V}_2), \ldots, \psi(v_i, \mathcal{V}_K)). \tag{7}$$

Here, $\psi(\cdot)$ represents the computation process of a given node with the corresponding anchor. Taking $\psi(v_i, \mathcal{V}_K)$ as an example, it is calculated as follows:

$$\psi(v_i, \mathcal{V}_K) = \min_{\mathcal{V}_j \in \mathcal{N}_{v_i}} \left( \mathcal{P}_{G_{sm}}(v_i, \mathcal{V}_j) \right) \tag{8}$$

where $\mathcal{V}_j \in \mathcal{N}_{v_i}$ denotes the existence of at least one path between node $v_i$ and anchor $\mathcal{V}_j$ in $\mathbf{G}_{sm}$. $\min(\cdot)$ denotes min-take operation.

After obtaining the SPD vectors of each node, combined with the output of the Transformer encoder, we reconstruct the awareness graph $\mathbf{G}_a$. This graph is used to re-measure the probability of a node establishing a link relationship. Specifically, take the node $v_i$ and $v_j$ as an example, we quantify this likelihood based on Eq. (9):

$$G_a(v_i, v_j) = \frac{1}{2}(z_i z_j^T + p_{v_j} p_{v_j}^T) * Rank \tag{9}$$

where $W_h$ is a learnable weight matrix, $||$ is a concatenation operation, and $z_i$ and $z_j$ are the output representations of Transformer encoder of node $v_i$ and $v_j$, respectively. $Rank$ is a PageRank matrix.

## Self-correlation fusion graph convolution framework

In this section, our objective is to acquire low-dimensional representations for nodes in both the similarity graph and perception graph, utilizing these representations for the identification of m6A modification sites. To achieve this, we propose the self-correlation fusion graph convolution framework, which comprises two main components: an self-correlation graph convolutional block for representation learning and a fusion block for integrating node representations from different graphs. The self-correlation graph convolutional block is designed to thoroughly capture the topological properties inherent in both the similarity and awareness graphs, independent of pre-existing node features. In contrast, the fusion block is utilized to integrate these low-dimensional node representations, which encapsulate topological information, thereby enhancing the quality of node representations for the subsequent classification task. Detailed descriptions of the framework are provided below.

We define the adjacent matrix $\mathbf{A} \in \mathbb{R}^{m \times m}$ for the similarity graph $G_{\mathrm{sm}}$, and define the node initial representation matrix as $\mathbf{Er} \in \mathbb{R}^{m \times d}$. Each value in the matrix is determined and iteratively optimized by the network. $\mathbf{A}$ and $\mathbf{Er}$ are input into a three-layer self-correlation graph neural networks to get the node embedding matrix $\mathbf{R}_{sm}$ of $G_{\mathrm{sm}}$:

$$
\begin{cases}
\mathbf{R}_{sm}^{(i+1)} = \mathbf{R}_{sm}^{(i)} + \zeta^{(i+1)} Gcov(\mathbf{A}, \mathbf{Er})^{(i+1)} \\
\zeta^{(i+1)} = \dfrac{1}{I+1}(i+1)
\end{cases}
\tag{10}
$$

where, $\zeta$ is the scaling superparameter to prevent the elements in the similarity matrix representation from becoming infinitesimal during graph convolution, and $I = 3$ denotes the number of convolution layers. $Gcov(\cdot)$ represents the convolution process. The hidden layer representation of layer $(i+1)$ and the representation of layer $(i)$ satisfies the following equation:

$$
H^{(l+1)} = \delta(\mathbf{A})^{-\frac{1}{2}}\left(\delta(\mathbf{A}) - \frac{1}{2}\left(\mathbf{A} + \mathbf{A}^T\right)\right)\delta(\mathbf{A})^{-\frac{1}{2}}H^{(l)}\mathbf{W}^{(l)}
\tag{11}
$$

where, $De_s$ denotes the degree matrix of the similar matrix $S$, and $diag(\cdot)$ denotes the diagonalization operation. $W$ is the learnable weight. The representation of the hidden layer is initialized to $Er$, that is, $H^{(0)} = \mathbf{Er}$. Similarly, we can compute the node embedding matrix for the awareness graph.

To attain a thorough understanding of RNA sequences, we employ a hybrid approach to integrate learned embedding representations. This methodology processes two distinct embeddings of the same sequence, addressing both local and global perspectives within the embedding space. We then establish weight relationships between the embeddings derived from similarity matrices and those obtained from sequence structure graphs. This is achieved through the implementation of specialized components: a local information extraction block and a global information extraction block. Each component is specifically designed to compute the weights associated with its respective embedding representation, thereby facilitating a more nuanced integration of local and global information.

Specifically, for a given similarity embedding matrix $\mathbf{M}^{sm} \in \mathbb{R}^{m \times d}$ and the corresponding awareness embedding matrix $\mathbf{M}^{aw} \in \mathbb{R}^{m \times d}$, we first obtain the weighted embedding matrix $\mathbf{M}^{we} \in \mathbb{R}^{m \times d}$ as:

$$\mathbf{M}^{we} = \mathbf{M}^{sm} + \mathbf{M}^{aw}. \tag{12}$$

We then enter $\mathbf{M}^{we}$ into the local and global information extraction blocks, respectively. For the local and global information extraction block, the extraction process is described as follows:

$$\begin{cases} \mathbf{M}^{we}_{lo} = f_{1D}\big(ReLU\big(f'_{1D}(\mathbf{M}^{we})\big)\big) \\ \mathbf{M}^{we}_{gl} = f_{1D}\big(\mathrm{ReLU}\big(f'_{1D}\delta(\mathbf{M}^{we})\big)\big) \end{cases} \tag{13}$$

where $\mathbf{M}^{we}_{lo}$ and $\mathbf{M}^{we}_{gl}$ denote the output of the local extraction block and that of the global extraction block respectively. $f_{1D}$ and $f'_{1D}$ represent a one-dimensional convolution layer containing normalized functions respectively. For the global information extraction block, we add the global average pooling layer $\delta$ on the basis of the local information extraction block. After that, we further obtain the fusion representation matrix as:

$$\begin{cases} \mathbf{M}^{we}_{final} = \mathrm{W}_p\mathbf{M}^{sm} + \big(I - \mathrm{W}_p\big)\mathbf{M}^{aw} \\ \mathrm{W}_p = \psi_{so}\Big(\mathbf{M}^{we}_{lo} + \mathbf{M}^{we}_{gl}\Big). \end{cases} \tag{14}$$

## Modification site identification

In this article, we use a support vector machine (SVM) to classify the final embedding matrix, a process represented by $p_i = \mathrm{SVM}\Big(\mathbf{M}^{we}_{final}\Big)$, where $p_i$ is a identification score. In classification problems, SVM identifies a hyperplane to divide data points of different classes, aiming to maximize the margin between the hyperplane and the support vectors for enhanced generalization performance. SVM's adaptability to high-dimensional spaces is a notable strength, achieved through the introduction of kernel functions that map data into higher-dimensional spaces, allowing the algorithm to address nonlinear classification problems. The algorithm excels in handling small sample sizes, high-dimensional datasets, and cases where data is not linearly separable.

## Metrics for experiments

For a binary classification problem, the prediction outcomes can be categorized into four distinct groups: true positive (TP), false positive (FP), true negative (TN), and false negative (FN). To comprehensively assess the predictive performance, we employ several key evaluation metrics, including accuracy (Acc), F1-score, precision (Prec), sensitivity (Sn), specificity (Sp), and Matthews correlation coefficient (MCC). Leveraging these metrics, we construct both the receiver-operating characteristic (ROC) curve and the precision-recall (PR) curve to visualize the prediction results. Additionally, we compute the area under the PR curve (AUPR) and the area under the ROC curve (AUC) to provide quantitative assessments of our model's performance. We evaluate our model using ten-fold cross-validation. Specifically, the dataset is divided into 10 approximately equal subsets, with each subset used as a validation set in turn, while the remaining nine subsets

are used for training. This process repeats 10 times. This method is particularly important for small datasets, as it allows each data point to be used for both training and validation, maximizing data utilization and reducing evaluation bias due to limited data.

## RESULTS

### Performance on A101 datasets

In this article, we conduct a comprehensive evaluation of our model through 10-fold cross-validation. Specifically, we partition the dataset into 10 folds, employing nine folds for training and reserving one fold for validation in each iteration. This process is repeated ten times to ensure the validation of each fold, with a crucial consideration that validation set data do not overlap with the training set data. Six metrics, including AUPR, AUC, Acc, F1 score, Prec, and Sen, are employed to quantify the model's performance. The results, detailed in Fig. 2 and Table 1, reveal promising outcomes. PR and ROC curves exhibit consistently good performance across all folds, with an average AUC of 91.96% and AUPR of 90.03%. This underscores the model's strong overall performance, further supported by an MCC value of 73.0%. Additionally, other indicators such as Acc, F1, Prec, and Sen consistently exceed 80%. Notably, the Acc value of 86.1% signifies superior recognition performance in positive samples, highlighting the model's efficacy in identifying potential positive instances. To emphasize the impact of structural perception, we visualize the scatter distribution of the validation set based on tSNE (*Van der Maaten & Hinton, 2008*), as depicted in Fig. 3. These experimental results demonstrate that the integration of structural information enhances the model's ability to recognize m6A modification sites.

### Performance comparison with other predictors

The comparative evaluation of three methods, namely M6AMRFS, BERMP, and RFAthM6A, is conducted through 10-fold cross-validation on the A101 dataset. Specifically, M6AMRFS utilizes two feature descriptors, namely dinucleotide binary coding and local site-specific dinucleotide frequency, to encode RNA sequences. It employs the F-score algorithm in conjunction with sequence forward search (SFS) to enhance feature representation and utilizes XGBoost (*Chen & Guestrin, 2016*) as a downstream classifier. BERMP utilizes gated recurrent units (GRU) (*Dey & Salem, 2017*) to represent RNA sequences and adopts an end-to-end training process for site identification. RFAthM6A attempts to classify various features from RNA sequences using machine learning methods. The comparison results, presented in Table 2 indicate that the proposed model outperforms the other three models across most indicators. The evaluation values for Acc and MCC reach 86.12% and 73.00%, respectively. This suggests the superiority of the proposed model. Notably, compared with the second method, the proposed model achieves a 5.07% improvement in Acc and a 0.45% improvement in MCC. M6AMRFS exhibits the lowest value in the MCC, while BERMP and RFAthM6A demonstrate similar values across all indicators. The experimental results affirm the effectiveness of the proposed model.

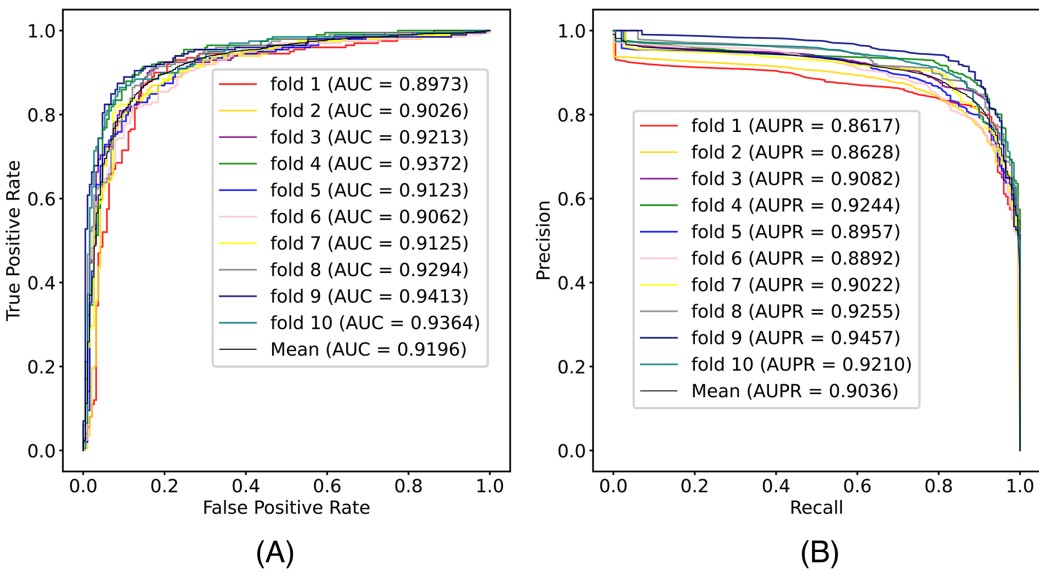

**Figure 2** The (A) ROC curves and (B) PR curves of M6ATMR on A101 dataset under 10-fold cross-validation.

## Ablation analysis

In this study, to assess the efficacy of the employed components, we introduce two additional variants and conduct validation using our model on the A101 dataset. The variants are labeled as M6A-SAI without Transformer (w/o Tr) and M6A-SAI without graph optimization (w/o Go). Specifically, w/o Tr employs K-mer representation in lieu of Transformer, while w/o Go omits the structure-aware graph optimization component. The comparison results, detailed in Fig. 4A, reveal a substantial decrease in all five indicators (AUC, AUPR, Acc, F1, and Prec) for both M6A-SAI and its variants. The degree of decrease is nearly identical for both variants, with the maximum decrease in AUC reaching 41.99% and the maximum decrease in Acc being 35.41%. This underscores the pivotal roles played by both the Transformer and the graph optimization process in the recognition task. To further underscore this point, we generated a scatter plot comparing our model with the two variants based on tSNE (*Van der Maaten & Hinton, 2008*). The results, displayed in Figs. 4B–4D, indicate that the scatter patterns of the two variants are nearly indistinguishable. This suggests the presence of a considerable number of false positive and false negative samples in the recognition results for both variants.

Furthermore, we discuss the individual contributions of each component. Specifically, we compare M6A-SAI with the model that solely utilizes fully connected and convolutional layers. We denote the model employing only fully connected layers as M6A-SAI-MLP, while the one utilizing graph convolutional layers is referred to as M6A-SAI-GCN. We conducted ten independent runs for M6A-SAI variants under different random seeds. As depicted in Table 3, Tables S1 and S2, we find that M6A-SAI-MLP exhibits inferior performance compared to both M6A-SAI and M6A-SAI-GCN on three datasets. It is probably because M6A-SAI-MLP solely extracts information from the feature matrix without capturing neighbor information adequately. Additionally, we

**Table 1 The value of some metrics in each fold.**

| Fold | AUC | AUPR | Acc | Sn | Sp | MCC | Prec | F1 |
|------|------|------|------|------|------|------|------|------|
| 1 | 0.897 | 0.862 | 0.861 | 0.865 | 0.857 | 0.722 | 0.865 | 0.865 |
| 2 | 0.903 | 0.863 | 0.846 | 0.814 | 0.879 | 0.694 | 0.876 | 0.844 |
| 3 | 0.921 | 0.908 | 0.851 | 0.819 | 0.884 | 0.704 | 0.881 | 0.849 |
| 4 | 0.937 | 0.924 | 0.887 | 0.894 | 0.879 | 0.774 | 0.886 | 0.890 |
| 5 | 0.912 | 0.896 | 0.848 | 0.824 | 0.874 | 0.701 | 0.872 | 0.848 |
| 6 | 0.906 | 0.889 | 0.828 | 0.829 | 0.826 | 0.720 | 0.833 | 0.831 |
| 7 | 0.913 | 0.902 | 0.859 | 0.834 | 0.884 | 0.719 | 0.883 | 0.858 |
| 8 | 0.929 | 0.926 | 0.866 | 0.854 | 0.879 | 0.733 | 0.881 | 0.867 |
| 9 | 0.941 | 0.946 | 0.887 | 0.859 | 0.916 | 0.775 | 0.914 | 0.886 |
| 10 | 0.936 | 0.921 | 0.879 | 0.869 | 0.889 | 0.758 | 0.892 | 0.880 |
| Mean | 0.920 | 0.904 | 0.861 | 0.846 | 0.877 | 0.730 | 0.878 | 0.862 |

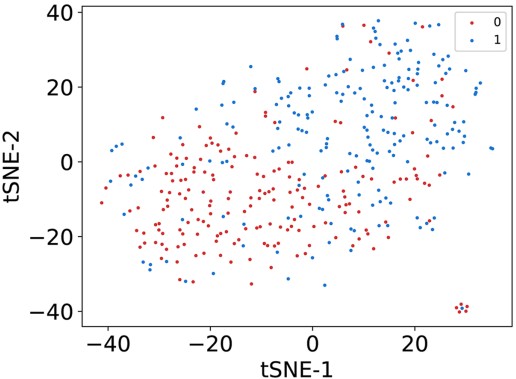

**Figure 3 The visualization results of sequence representation.**

**Table 2 The comparison results of different recognition methods.** N.A. denotes the value of the indicator is not provided by corresponding studies.

| Models | Acc | MCC | Sn | Sp |
|--------|------|------|------|------|
| M6A-SAI | 86.12 | 73.00 | 84.61 | 87.67 |
| M6AMRFS | 81.05 | 62.10 | 80.67 | 81.43 |
| BERMP | N.A. | 72.60 | 82.30 | 90.00 |
| RFAthM6A | N.A. | 72.55 | 82.22 | 90.00 |

observe a significantly superior performance of M6A-SAI over M6A-SAI-GCN. Although M6A-SAI-GCN can learn graph structure information, it is susceptible to noise present in the initial graph which hampers effective information aggregation. Conversely, by incorporating a graph optimization mechanism, M6A-SAI filters out this noise thereby enhancing node representation and improving prediction performance.

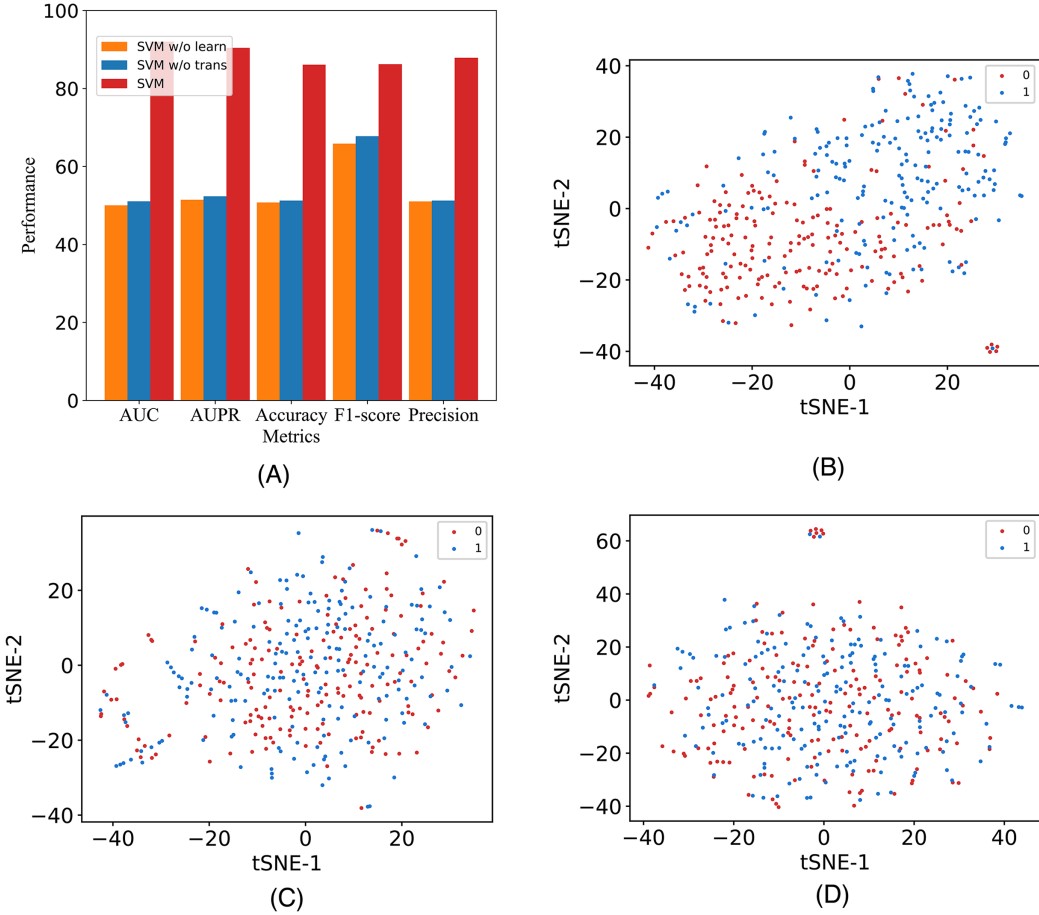

**Figure 4 The metrics and the scatter visualization results of ablation results.** (A) The metrics of M6A-SAI and variants. (B) Visualization of M6A-SAI. (C) Visualization of M6A-SAI without Go. (D) Visualization of M6A-SAI without Tr.

**Table 3 The metrics of M6A-SAI and its variants.**

| Models | Acc | MCC | AUC | AUPR |
|---|---|---|---|---|
| M6A-SAI | 0.861 | 0.730 | 0.920 | 0.904 |
| M6A-SAI-MLP | 0.522 | 0.387 | 0.514 | 0.523 |
| M6A-SAI-GCN | 0.554 | 0.412 | 0.589 | 0.534 |

## Analysis of the number of layers in graph neural networks

In this study, we establish the number of network layers for the graph convolution as 4. To elucidate the rationale behind this decision, we construct variants of the model with different numbers of network layers in the graph convolution as the backbone network. The comparative results of these variants with the model are depicted in Fig. 5. It is noteworthy that by considering the variant with four network layers as the critical point, we observe that when the number of layers is less than four, the model's performance metrics increase with the growing number of network layers. However, when the number

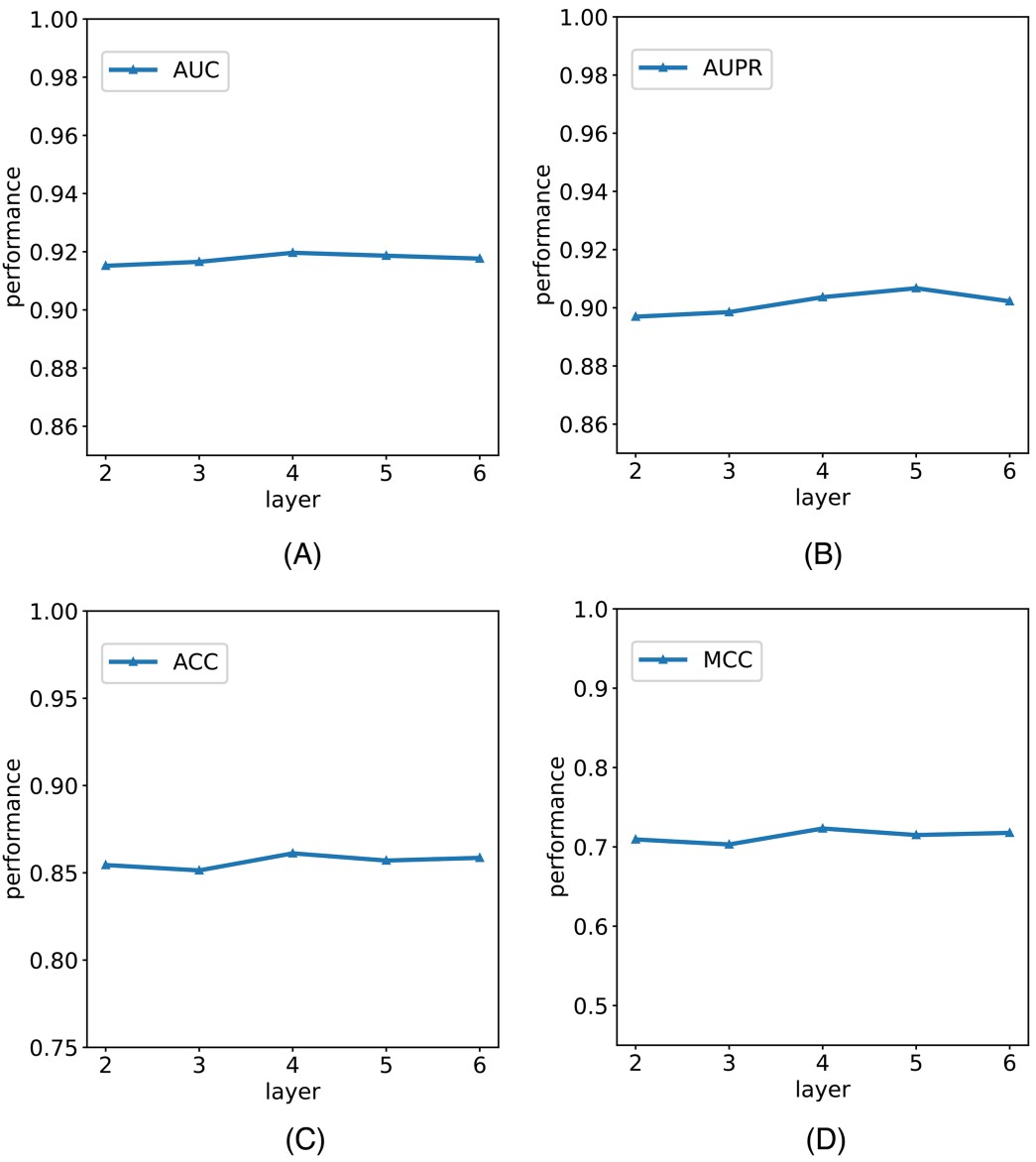

**Figure 5 The comparison results of M6A-SAI and its variants with different numbers of network layers.** (A) AUC values on different layer. (B) AUPR values on different layer. (C) ACC values on different layer. (D) MCC values on different layer.

of layers exceeds four, the model's metrics begin to decline with an increasing number of layers. Notably, with six layers, the model registers a reduction in AUC. We posit that this phenomenon is attributable to the initial stages, where augmenting the number of network layers facilitates the model in learning more intricate and abstract features, thereby enhancing its ability to represent data. However, as the number of network layers continues to increase, the model's parameters also escalate, amplifying the risk of overfitting. In scenarios with limited training data and an excessively complex model, this may lead to favorable performance on the training set but poorer generalization to unseen

data. Consequently, selecting an optimal number of network layers is crucial to strike a balance between feature representation capacity and the risk of overfitting.

## Analysis of the type of graph neural networks

In our model, we adopt the simplest graph convolutional network (GCN) as the backbone network, primarily for the sake of model simplification. To emphasize this choice, we further explored three variants of graph neural networks (GNNs), including ChebNet (*Tang, Li & Yu, 2019*), graph attention network (GAT) (*Velickovic et al., 2017*), and GraphSAGE (*Hamilton, Ying & Leskovec, 2017*), as potential replacements for GCN in the backbone network. The comparison results, illustrated in Fig. 6 and detailed in Table 4, reveal that while different variants exhibit certain effects within our model, none outperforms GCN across all metrics. It is noteworthy that the other variants generally employ more intricate network structures, such as GAT, which introduces an attention mechanism, thereby increasing computational costs without yielding significant improvements in results. Despite their complexity, these variants do not surpass the performance of the simpler GCN in the context of m6A modification site identification. In summary, the results indicate that GCN stands out as the most suitable GNN type for our model in the task of m6A modification site identification, striking a balance between effectiveness and model simplicity.

## Performance on independent data

To evaluate the generalization M6A-SAI, we adopt the datasets with different data size from different species. Specifically, we compare M6A-SAI with M6AMRFS, BERMP, RFAthM6A, AdaRM (*Song et al., 2023*), iRNA-m6A (*Dao et al., 2020*) and TS-m6A-DL (*Abbas et al., 2021*). Table 5 demonstrates that M6A-SAI outperforms BERMP (the second) on human-liver dataset, achieving an Acc that is 0.9% higher, MCC that is 3% higher, Sn that is 1.2% higher and Sp that is 3.9% higher. Table 6 demonstrates that M6A-SAI outperforms TS-m6A-DL (the second) on mouse-heart dataset, achieving an Acc that is 3.7% higher, Mcc that is 7.4% higher, Sn that is 2.9% higher, and Sp that is 4.6% higher. Besides, we notice that BERMP achieves satisfactory prediction performance on mouse data but worse performance on human-liver dataset, but M6A-SAI exhibits remarkable performance on all datasets, which illustrate M6A-SAI has good generalization compared with other methods.

## Analysis of the overfitting of M6A-SAI

To illustrate the effectiveness of M6A-SAI in an intuitive fashion, we visualize learning curve to validate that there is overfitting in training process. As shown in Fig. 7, we find that with the increase in sample size, the performance of M6A-SAI on the test set exhibits gradual improvement and eventually stabilizes. Furthermore, there is no significant disparity in performance between the M6A-SAI trained on the training set and tested on the test set, indicating that despite operating on a small dataset, there is not overfitting in training process.

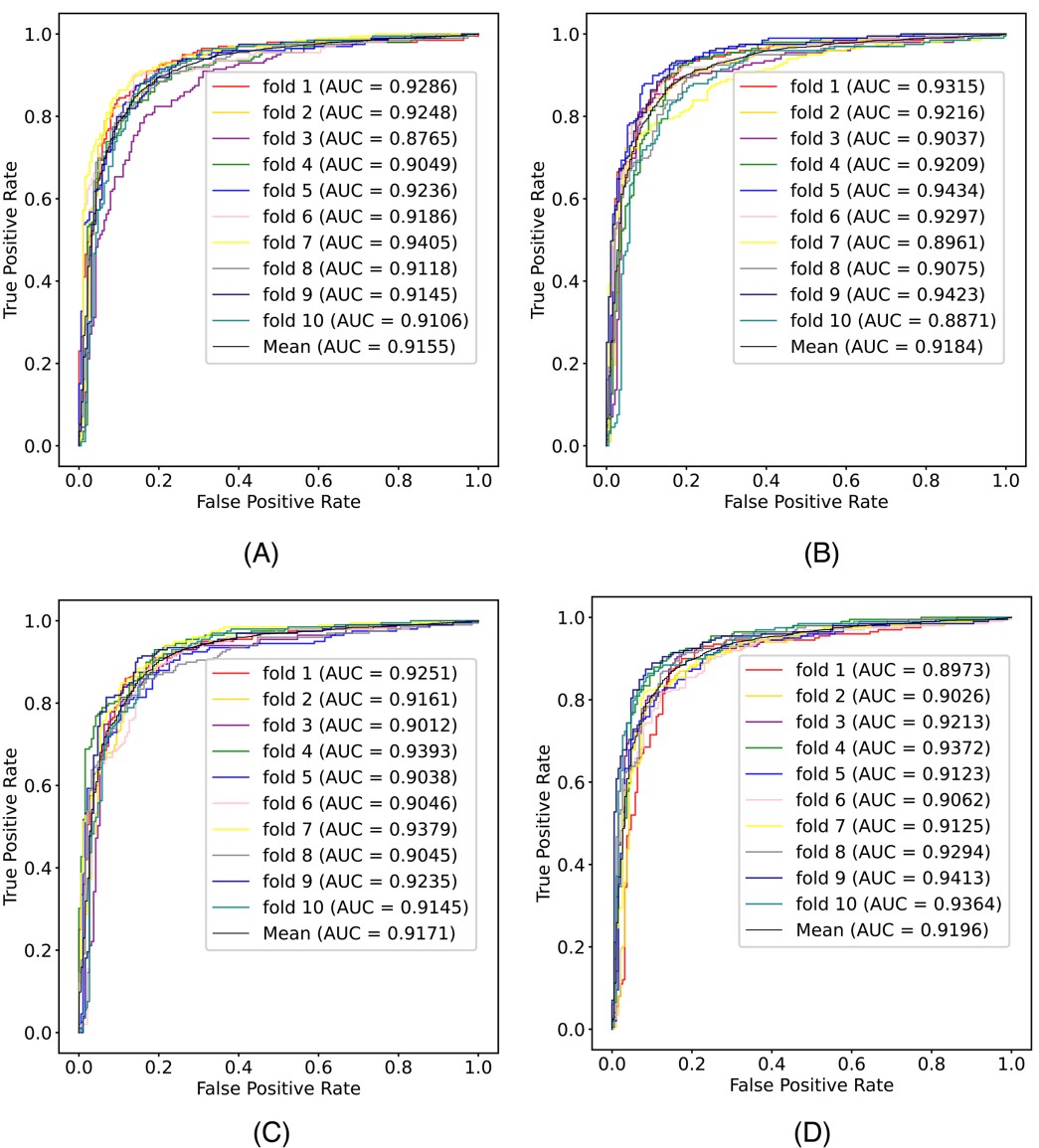

**Figure 6 The comparison results of M6A-SAI and its variants with different network types.**
(A) ChebNET. (B) GAT. (C) GraphSAGE. (D) M6A-SAI.

**Table 4 The detailed metrics of M6A-SAI and its variants with different network types.**

| GNN | AUC | AUPR | Acc | Sn | Sp | MCC | Prec | F1 |
|---|---|---|---|---|---|---|---|---|
| ChebNet | 0.915 | 0.900 | 0.851 | 0.852 | 0.850 | 0.703 | 0.857 | 0.854 |
| GAT | 0.918 | 0.906 | 0.853 | 0.847 | 0.860 | 0.707 | 0.865 | 0.856 |
| GraphSAGE | 0.917 | 0.902 | 0.853 | 0.860 | 0.846 | 0.707 | 0.855 | 0.857 |
| M6A-SAI | 0.920 | 0.904 | 0.861 | 0.846 | 0.877 | 0.723 | 0.878 | 0.862 |

**Table 5 The metrics of M6A-SAI and other baseline model under human-liver dataset.**

| Models | Acc | MCC | Sn | Sp |
|---|---|---|---|---|
| M6A-SAI | 0.820 | 0.640 | 0.832 | 0.808 |
| M6AMRFS | 0.788 | 0.580 | 0.769 | 0.777 |
| BERMP | 0.811 | 0.610 | 0.820 | 0.769 |
| RFAthM6A | 0.790 | 0.599 | 0.812 | 0.811 |
| AdaRM | 0.803 | 0.606 | 0.822 | 0.783 |
| iRNA-m6A | 0.790 | 0.580 | 0.782 | 0.799 |
| TS-m6A-DL | 0.805 | 0.611 | 0.820 | 0.790 |

**Table 6 The metrics of M6A-SAI and other baseline model under mouse-heart dataset.**

| Models | Acc | MCC | Sn | Sp |
|---|---|---|---|---|
| M6A-SAI | 0.787 | 0.576 | 0.822 | 0.753 |
| M6AMRFS | 0.701 | 0.428 | 0.700 | 0.717 |
| BERMP | 0.732 | 0.475 | 0.789 | 0.698 |
| RFAthM6A | 0.724 | 0.441 | 0.732 | 0.743 |
| AdaRM | 0.739 | 0.483 | 0.802 | 0.677 |
| iRNA-m6A | 0.713 | 0.430 | 0.705 | 0.721 |
| TS-m6A-DL | 0.750 | 0.502 | 0.793 | 0.707 |

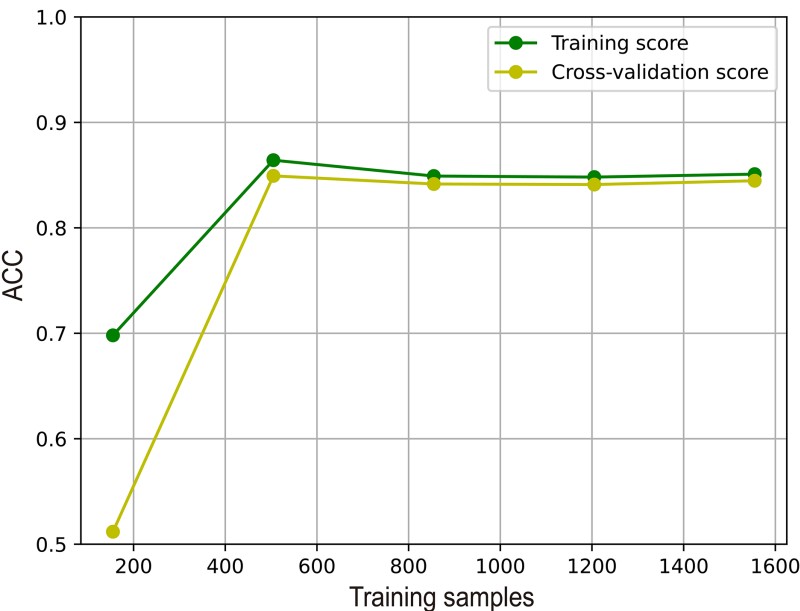

**Figure 7 The learning curve of M6A-SAI in the training process.**

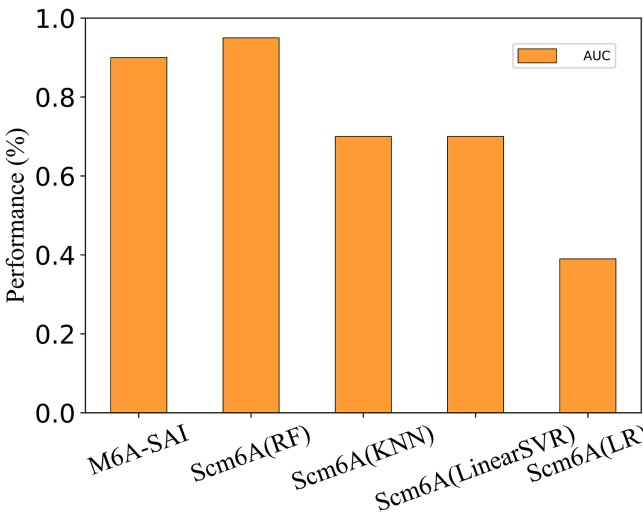

**Figure 8  The metrics of M6A-SAI and Scm6A under single-cell dataset.**

## Performance on single-cell datasets

To further evaluate the generalization performance of M6A-SAI, we apply M6A-SAI to single-cell datasets. Despite the original design of M6A-SAI not being intended for such data, we believe that M6A-SAI can still be utilized for predicting m6A modification sites at the single-cell level. Specifically, we employ the dataset provided by Scm6A and followed the data processing approach of Scm6A to obtain the model input. Subsequently, M6A-SAI is employed for predicting m6A modification sites. As shown in Fig. 8, although M6A-SAI does not surpass Scm6A (RF) in terms of AUC, it outperforms Scm6A (KNN), Scm6A (LinearSVR), and Scm6A (LR) significantly. It is probably because that M6A-SAI is not originally designed for single-cell data, which often contains numerous sparse values, impacting the performance of M6A-SAI.

## DISCUSSION AND CONCLUSION

In this article, we propose a structure-aware m6A prediction framework, M6A-SAI, designed to achieve precise predictions of m6A modification sites. M6A-SAI integrates Transformers and graph convolutional networks to perform representation learning on sequences from a graph perspective. During this process, the structural awareness of the graph is introduced to extend sequence similarity information to a more comprehensive graph structure, enabling finer-grained learning. This includes acquiring the low-dimensional representation of nodes based on graph information and obtaining a comprehensive node representation through fusion algorithms. Extensive experiments underscore the effectiveness of the proposed strategy. Ablation analysis and backbone network evaluation reveal that the current structure of M6A-SAI is an optimal choice. The optimization of each component, particularly the graph structure, proves pivotal in enhancing the model's efficacy. Comparative analyses with other methods further highlight the accuracy and robustness of our model. In conclusion, M6A-SAI introduces structural insights from graphs to sequence representation and classification tasks, serving

as an effective complement to existing m6A modification site prediction tasks. However, we acknowledge certain limitations in our study. First, the strategy is not validated on a broader range of species and tissue types, limiting its potential biological interpretability. Second, due to data limitations, the proposed model does not execute relevant recognition strategies on multi-scale RNA sequences. These issues will be addressed in future work. Third, more negative sample selection strategies are not explained further. Addressing these issues will be a key focus of our future work. In future work, we plan to explore different negative sample selection strategies and conduct comparative experiments to assess their impact on model generalization, particularly in terms of generalization across different biological contexts. Moreover, we acknowledge the potential for integrating our method with long-read sequencing data in future work to further enhance its applicability and performance.

### Funding
This work was supported by Analysis and monitoring of spatial dynamic change and environmental impact of Three Gorges Reservoir Impoundment Operation on the middle and lower reaches of Yangtze River shoreline. The funders had no role in study design, data collection and analysis, decision to publish, or preparation of the manuscript.

### Grant Disclosures
The following grant information was disclosed by the authors:
Three Gorges Reservoir Impoundment Operation on the Middle and Lower Reaches of Yangtze River Shoreline.

### Competing Interests
The authors declare that they have no competing interests.

### Author Contributions
- Yue Yu conceived and designed the experiments, performed the experiments, analyzed the data, prepared figures and/or tables, authored or reviewed drafts of the article, and approved the final draft.
- Shuang Xiang analyzed the data, prepared figures and/or tables, and approved the final draft.
- Minghao Wu conceived and designed the experiments, prepared figures and/or tables, authored or reviewed drafts of the article, and approved the final draft.

### Data Availability
The raw data and code are available in the Supplemental Files.

## Supplemental Information

Supplemental information for this article can be found online at http://dx.doi.org/10.7717/peerj.18878#supplemental-information.

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
