# Peer review of "Injecting structure-aware insights for the learning of RNA sequence representations to identify m6A modification sites"

_PeerJ, doi:10.7717/peerj.18878_

## Round 0.1 · original submission · Major Revisions

Dear authors,

We kindly request that you carefully review the comments provided by the reviewers. Their valuable suggestions offer insights to enhance your manuscript. Incorporate their suggestions and carefully address all comments in your manuscript; it will significantly strengthen its content. Thanks

Reviewer 1 ·

Basic reporting

The machine learning methods are well explained in some detail, but there is missing information regarding the processing of the data itself.
* It reads that "validation set data do not overlap with the training set data" - more details are required regarding how this is done as it is very important for showing there is no overfitting here (particularly relevant given the small dataset size).
* How are the negatives chosen in this data?
* What is the chosen sequence lengths used by the model (and how did the authors optimise this aspect)?

Experimental design

The authors are using complex model architectures (transformers, graph convolutional networks) for what is actually a small dataset. More justification is needed for using this specific dataset, and what measures/checks have been done to be sure that overfitting has not occurred. Furthermore, how does a simpler model fare (e.g. full connected/convolutonal layers only, with the data/all other things being held the same)?

Validity of the findings

Figure 4 is confusing and I wonder if it could be checked - removing either of the transformer or graph parts reduces the model to almost random guessing as per the AUC. Yet, the other metrics, particularly the F1-score, show that the model is performing well (albeit not as good as the one with both layers included).

Reviewer 2 ·

Basic reporting

The authors proposed a new m6A modification preodictor named M6A-SAI by infusing structure-aware insights into sequence representation learning based on deep learning and SVM algorithms. They evaluated the performance of M6A-SAI and its variants with different number of network layers, and performed the performance comparsion between MA6-SAI and other m6A predictors.

Experimental design

The performance of M6A-SAI was comparable to other tested m6A predictors including BERMP and RFAthM6A. In addition, the performance of M6A-SAI was not fully evaluated with multiple m6A datasets from different species.

Validity of the findings

The novelty of this manuscript was not clearly discribed. Although the complexity of prediction algorithm used in M6A-SAI, the performance of m6A prediction was not significantly improved.

·

Basic reporting

Yes. I think this paper is clear and unambiguous, professional English used throughout. But, literature references, sufficient field background/context provided not good enough.

Experimental design

Experimental design is OK

Validity of the findings

Validity of the findings is OK

Additional comments

1. The proposed framework is quite complex, involving multiple components such as the Transformer encoder, similarity graph, structure-aware optimization block, and self-correlation fusion graph convolution framework, which may increase implementation and comprehension difficulty.
2. Long-read direct sequencing already performs well in identifying m6A, so this method may not have significant future applicability.
3. It is recommended that the authors compare their method with the following single-cell m6A methods.10.1093/gpbjnl/qzae039

Reviewer 4 ·

Basic reporting

The paper presents a deep learning architecture to predict RNA m6A modification sites. It provides extensive experimental results and claims that M6A-SAI outperforms existing prediction methods. Addressing these suggestions would further strengthen the manuscript.
(1) English expressions need to be edited more careful and more native, in this manuscript, there are some mistakes. For example: Line 74
(2) The abstract necessitates revision. It is advisable to address the following key components in the revised abstract: Question or problem under investigation. Basic methodology (without delving into specific details).
(3) The readability of your ideas would greatly benefit if you provide a reader with explanations for all notations. This is especially crucial when it comes to procedure listings.
(4) The introduction requires revision and modification. I suggest revising it with consideration for the following points: context and motivation, the problem at hand, overview of related previous works, including their limitations, list of contributions and key results, organization of the subsequent sections.
(5) I suggest that you define acronyms when you first mention them in the text, and then provide explanations to ensure clarity and understanding. This approach will make the text more organized and easier to read, making it more accessible for readers.
(6) The manuscript would benefit from a clearer explanation of how this methodology differs fundamentally from existing Transformer approaches used in similar contexts.
(7) The mathematical formulas are not displaying correctly, and the fonts appear to be inconsistent. The authors should review the formatting.

Experimental design

See above

Validity of the findings

See above

Additional comments

See above

---

## Round 0.2 · Major Revisions

Dear authors,

We kindly request that you carefully review the comments provided by the reviewers. Their valuable suggestions offer insights to enhance your manuscript. Incorporate their suggestions and carefully address all comments in your manuscript; it will significantly strengthen its content. Thanks

Reviewer 1 ·

Basic reporting

Abstract - first sentence is incorrect. Should be perhaps "it is crucial to accurately identify its modification sites on RNA sequences"

Abstract - sentence two "Traditional approaches to m6A modification site identification primarily focus on" should perhaps be "Traditional machine learning based approaches to [...]" since otherwise this would be misunderstood as general modification detection via experiments.

Limitations section is now needing updating since the authors actually have now tested on different species.

New line 303: "This approach generally helps to improve dataset quality." Use of flanking sequence provides a feature set, but it is not clear what is meant here about it improving dataset quality.

Experimental design

I am somewhat unsatisfied regarding the answers provided to my previous questions regarding the study design, and therefore follow up on the authors responses below:

* When the authors write cross validation is used to ensure no overlaps, is it simply meant that the same m6A coordinate is not present in both training and validation sets for the same model? But is it possible that the 101bp sequences (that is when including the flanking sequence) do overlap between the two sets? This is valid to address given that m6A sites often occur at nearby coordinates on the same mRNAs, and overlapping features can have an impact on potential overfitting.

* For negatives the authors reply that they have selected adenines, but what is the procedure for selecting these As: for example, are these randomly picked from the transcriptome, randomly from the same genes as the positives, and also do they align with the DRACH motif? I have seen a lot of different approaches taken in the literature, and these decisions impact how the model should be interpreted. If the negatives are random non-DRACH adenosines, then how would this affect the generalisability in making downstream predictions? This is an important consideration given that the vast majority of adenosines in the transcriptome are unmethylated, whilst the training data used was balanced at a 1:1 ratio.

* The authors have chosen 101 nt but did they also try out other lengths (given that transformers are designed to capture highly long range contexts). And perhaps they can elaborate on why only 41 nucleotides (+/- 20bp) were considered for the additional datasets (which have been added since the last version)?

Validity of the findings

I still do not understand how the authors obtain an AUC of 0.500 for the MLP - please explain more specifically what is meant by this statement "It's probably because M6A-SAI-MLP solely extracts information from the feature matrix without capturing neighbor information adequately." as surely at the very least an MLP should be able to learn some information about DRACH being associated with m6A deposition in the positives?

Additional comments

In the revised version there is a paragraph 294-306 outlining the main contribution of the article, which is the use of the advanced architectures towards the problem of m6A prediction. Since the (complex) modelling strategy is very much the focus, there should be code provided for others to be able to apply such methods to their data.

Reviewer 4 ·

Basic reporting

no comment

Experimental design

no comment

Validity of the findings

no comment

Additional comments

no comment

---

## Round 0.3 · Minor Revisions

Dear authors,

Please address the final minor comment about the MLP model (i.e. check that it is correct)

Reviewer 1 ·

Basic reporting

The authors have improved the clarity of their explanations in light of the points I raised previously.

Experimental design

no comment

Validity of the findings

I'm still a little surprised that "M6A-SAI-MLP" has no information whatsoever in distinguishing positives and negatives (AUC=0.5, i.e. no better than random), although I agree with the others that it would not be expected to perform as well as the other approaches. I was wondering if the authors had checked this method using some of the other datasets, just to be absolutely sure. But otherwise no further comment.

---

## Round 0.4 · accepted · Accept

The authors introduced M6A-SAI, showcasing its effectiveness as a reliable method for identifying RNA m6A modification sites. The current revised version provides a clear and detailed explanation of the proposed identification methods, emphasizing their robustness. M6A-SAI proves to be highly beneficial in advancing RNA biology.